# Machine Learning Approach to Understand Worsening Renal Function in Acute Heart Failure

**DOI:** 10.3390/biom12111616

**Published:** 2022-11-02

**Authors:** Szymon Urban, Mikołaj Błaziak, Maksym Jura, Gracjan Iwanek, Barbara Ponikowska, Jolanta Horudko, Agnieszka Siennicka, Petr Berka, Jan Biegus, Piotr Ponikowski, Robert Zymliński

**Affiliations:** 1Institute of Heart Diseases, Wroclaw Medical University, 50-556 Wroclaw, Poland; 2Department of Physiology and Patophysiology, Wroclaw Medical University, 50-368 Wroclaw, Poland; 3Institute of Heart Diseases, Student Scientific Organization, Wroclaw Medical University, 50-368 Wroclaw, Poland; 4Faculty of Electrical Engineering, Warsaw University of Technology, 00-614 Warszawa, Poland; 5Department of Information and Knowledge Engineering, Prague University of Economics and Business, W. Churchill Sq. 1938/4, 130 67 Prague, Czech Republic

**Keywords:** acute heart failure, machine learning, clustering, artificial intelligence, cardiorenal syndrome

## Abstract

Acute heart failure (AHF) is a common and severe condition with a poor prognosis. Its course is often complicated by worsening renal function (WRF), exacerbating the outcome. The population of AHF patients experiencing WRF is heterogenous, and some novel possibilities for its analysis have recently emerged. Clustering is a machine learning (ML) technique that divides the population into distinct subgroups based on the similarity of cases (patients). Given that, we decided to use clustering to find subgroups inside the AHF population that differ in terms of WRF occurrence. We evaluated data from the three hundred and twelve AHF patients hospitalized in our institution who had creatinine assessed four times during hospitalization. Eighty-six variables evaluated at admission were included in the analysis. The k-medoids algorithm was used for clustering, and the quality of the procedure was judged by the Davies–Bouldin index. Three clinically and prognostically different clusters were distinguished. The groups had significantly (*p* = 0.004) different incidences of WRF. Inside the AHF population, we successfully discovered that three groups varied in renal prognosis. Our results provide novel insight into the AHF and WRF interplay and can be valuable for future trial construction and more tailored treatment.

## 1. Introduction

Acute heart failure (AHF) remains a significant problem with a high mortality and a massive financial burden for healthcare providers [1,2]. AHF is a multidimensional state with a complex interplay between the cardiovascular and other systems, including the renal. The pathological condition of simultaneous dysfunction of the kidneys and heart, in which the disorder of one organ induces the damage of the second one, is called cardiorenal syndrome [3]. One of the clinical manifestations of cardiorenal syndrome is the worsening renal function (WRF), which can be defined as, e.g., an increase in serum creatinine or/and a decrease in urine output in a specified period [4]. WRF is a frequent complication overlapping the AHF, especially in conditions of intensive cardiac care units [5], and is associated with prolonged hospitalization and diminished survival [4]. The population of AHF patients endangered by the WRF is heterogenous, and so is the postulated WRF’s impact on prognosis. Some authors showed contrary evidence that WRF has a negative, neutral, or even positive effect [4,6,7]. Considering this uncertainty, we presumed that the current lack of well-established classifications describing the risk of WRF is insufficient and does not reflect significant clinical differences between AHF patients. Thus, we decided to analyse the heterogeneity of the AHF population by resorting to novel methods of data analysis, aiming to describe different risk groups of WRF and, further, its impact on prognosis. Importantly, we have only included variables, which are the standard-of-care parameters routinely assessed during AHF patient monitoring.

Data science algorithms, especially Machine Learning (ML), enable novel, clinically important insight into existing data and distinguish previously unrecognized patterns [8]. Clustering is an unsupervised ML technique that organizes the set of data into internally similar subgroups. We presumed that this technique, which was successfully leveraged in marketing [9], could as well prove its value in cardiovascular research. Considering these advances, we decided to implement clustering in the AHF population to understand the occurrence and significance of the WRF better.

## 2. Materials and Methods

### 2.1. Study Population

We have retrospectively analysed three hundred and twelve acute heart failure (AHF) patients from two registries conducted in our institution between 2010–2012 and 2016–2017. Our previous papers described the eligibility criteria in both registries [10]. Heart failure diagnosis was stated according to the current ESC guidelines by a responsible physician [11,12]. To ensure the creatinine course in every patient and avoid missing values in the analysis, we have only included the patients who had serum creatinine assessed at four points, i.e., at admission, after 24 and 48 h of hospitalization, and at discharge.

### 2.2. Worsening of the Renal Function Evaluation

As there was a significant lack of data about diuresis and GFR or parameters indispensable for its calculation, we have based the diagnosis of worsening renal function (WRF) and acute kidney injury (AKI) on creatinine assessment only. AKI was defined according to the KDIGO guidelines as the ≥0.3 mg/dL increase of serum creatinine in 48 h [13]. WRF was defined as the ≥0.3 mg/dL increase of serum creatinine at any point during hospitalization. We decided to analyse both of these phenomena in order to caption as many renal endpoints as possible. Throughout the paper we will stick to using the term WRF, as it is a broader qualification.

### 2.3. Clustering and Data Analysis

Variables included in the analysis are shown in Table 1. Initially, we chose 86 variables regarding the patient’s clinical status, i.e., HF subtype, aetiology, comorbidities, symptomatology, and biochemical presentation. All parameters were assessed at patient admission to the hospital. Variables were manually screened to eliminate potential errors; e.g., anomalies, single values out of range, etc. The dataset was implemented into RapidMiner and autocleaning was performed. Variables with over 90% stability, 10% of missing values, or correlated with at least r = 0.6 were meant to be removed, but none of the variables fulfilled these criteria. Missing values were replaced by average values, as clustering algorithms cannot proceed with missing values. Further, nominal values were converted into numerical, and all the numerical parameters were normalized to range from 0 to 1, so each variable had the same impact on the calculated distance.

Clustering is a widely used descriptive data analysis method on the border between statistical analysis and data mining with a relatively long history. The goal of clustering (also called segmentation) is to identify groups of similar examples. Thus, the critical issue in clustering is a proper definition of similarity or distance. There are several clustering methods and algorithms that can be divided into various types, such as hierarchical versus partitional, exclusive versus overlapping versus fuzzy, and complete versus partial [14].

We used the k-medoids algorithm in our experiments. K-medoids is a partitional method that creates non-overlapping clusters. The number of resulting groups must be specified in advance. The algorithm repeatedly re-assigns the examples into the given number of clusters by minimizing their distance to a centroid and recomputes the centroids. Unlike k-means clustering, where cluster centroids are computed by averaging values for examples in a given cluster, each cluster in k-medoids clustering is represented using an existing, most representative example. This makes the results of the k-medoids clustering easier to interpret. The implementation in RapidMiner luckily offers the option to tune hyperparameters of the algorithm automatically. In our case, we adjusted the number of clusters and the similarity measure. The process of the clusters’ calculation performed in RapidMiner is displayed in Figure 1, and the file is attached in Appendix A.

We assessed the quality of clustering using the Davies–Bouldin index [15]. This index evaluates the quality of clustering considering the intra-cluster distance (that should be low) and inter-cluster distance (that should be high). The lower the value of the Davies–Bouldin index, the better the clustering.

Associations between clusters and clinical variables were evaluated. The normality was checked using K-S, Shapiro–Wilk, and Lilliefors tests. Parameters with normal distributions are shown as means ± standard deviations. The non-normal variables are displayed as the medians and interquartile ranges. Categorical variables are shown as numbers and percentages (Table 2). Statistical significance was evaluated using analysis of variance; the *p* below 0.05 was considered statistically significant. Clustering was performed in RapidMiner 9.1 (RapidMiner GmbH, Dortmund, Germany), and the statistical assessment was conducted in STATISTICA 12 (StatSoft Polska Sp. z o.o., Krakow, Poland).

## 3. Results

### 3.1. Population Characteristics

The population consisted of 312 patients, predominantly men (75%). The mean age was 68.4 ± 13.054. Average values of important clinical parameters were systolic blood pressure-130 mmHg (110–150), ejection fraction–32% (25–45), NT-proBNP–5659 pg/mL (3119–10,572), and serum creatinine −1.21 mg/dL (1.005–1.49). A detailed description of the patient’s characteristics, including characteristics by clusters, is displayed in Table 2.

### 3.2. Clustering

The population was segmented into three clusters, enumerated from 0 to 2. Groups included were, respectively, 158, 110, and 44 patients.

#### 3.2.1. Cluster 0

Cluster 0 was the most numerous one. It comprised the highest proportion of chronic HF with reduced ejection fraction, with the underlying cause of coronary artery disease. Patients usually had a history of PCI/CABG and electrical device implantation. COPD and insulin-dependent diabetes were most frequently reported. Clinical status comprised common pulmonary congestion, moderate limb oedema, and the lowest heart rate. In laboratory parameters, they presented the lowest Ast, Alt, ferritin, IL-6, and NT-proBNP.

#### 3.2.2. Cluster 1

Among other clusters, this group was composed predominantly of older women. They manifested the first manifestation of HF, with preserved ejection fraction and high comorbidity burden, i.a., diabetes and hypertension. Their clinical presentation was reflected by the most frequent NYHA IV, least frequent lower limb oedema and pulmonary congestion, and highest blood pressure. In laboratory measurements, they reached the lowest haemoglobin, HCO3, bilirubin, GGTP, and the highest serum sodium and potassium concentration, serum osmolarity, glucose, Ast, Alt, IL-6, and ferritin.

#### 3.2.3. Cluster 2

The last group was the youngest, with the highest proportion of males and the lowest ejection fraction. They reported the highest stroke history and presented with frequent ascites and hepatomegaly. They achieved the highest HGB, HCT, MCV, bilirubin, GGTP, Fe, NT-proBNP, urine creatinine, and urea and the lowest albumin in laboratory parameters. They were also the most frequent active alcohol users and smokers.

The most important clinical features of each cluster are shown in Table 3 and Figure 2.

### 3.3. Outcome

The global one-year mortality in the studied group was 24% (74 events occurred). The mortality did not significantly differ between the clusters (*p* = 0.2), from cluster 0 to cluster 2: 22% vs. 22% vs. 34%. The Cox regression was performed, but none of the cluster’s hazard ratios reached statistical significance (*p* = 0.35, *p* = 0.75, *p* = 0.0.09), and neither did the Kaplan–Meier estimation (*p* = 0.21).

Clusters differed in terms of the time of hospitalization, AKI, and WRF occurrence. Patients in cluster 2 were the least likely to develop AKI or WRF and were hospitalized for the longest time.

The outcomes and findings are summarised in Table 4.

## 4. Discussion

The WRF and AKI in AHF are common complications associated with ominous outcomes [4]. The occurrence of AKI has been estimated at 9–13% of AHF patients [16,17]. The underlying causes of the WRF in AHF are complex and not fully understood; the most prominent hypotheses include the impact of, i.a., congestion [18]. Given this lack of specific evidence, we decided to analyse the heterogeneity of the AHF population in the context of WRF occurrence and possible clinical phenotypes which determine it.

The ML-based analysis is gaining popularity in cardiovascular research [19]. There were some magnificent attempts to implement ML in the HF population [20,21,22,23,24,25,26]. Yagi tried to identify distinct phenotypes among AHF patients who experienced WRF [27]. Nevertheless, our study is the first to incorporate clustering into the analysis of the HF population, aiming to distinguish subgroups varied in terms of the WRF. The clustering techniques were able to distinguish three interesting clinical subtypes with different pathophysiology and implications for the outcome.

### 4.1. Cluster 0

This cluster represents the population of older men with chronic HF. We can assume that these patients represent the population with a relatively long history of cardiovascular treatment as they are frequently secured with the electric device and have undergone coronary intervention. They have also been saddled with comorbidities, i.e., end-stage insulin-dependent diabetes and COPD. As these patients represent the group of the chronic and fragile population, therapeutic interventions should be targeted at stable heart failure and comorbidities management [28,29,30].

### 4.2. Cluster 1

Cluster 1 is mainly composed of females. It is the oldest population with the first manifestation of HF, non-ischaemic aetiology, and preserved ejection fraction. They present signs of minimal congestion. In the biochemical assessment, patients in cluster 1 reached the highest serum creatinine, sodium potassium, and osmolarity. This phenotype corresponds with the described HFpEF phenotype [31]. Cluster 1 achieved the highest concentration of selected inflammatory biomarkers (IL-6, ferritin), and high activation of inflammatory pathways was reported to be unique for the HFpEF [32]. Recent studies showed that higher osmolarity correlates with the incidence of WRF in AHF [33]. Importantly, this group reached the highest incidence of AKI and WRF but moderate mortality; our consideration of its explanation is presented in the next paragraph. As the HFpEF population currently suffers from the lack of evidence-based treatment, therapeutic interventions should focus on comorbidities management and lifestyle changes [2]. Some hope for efficient pharmacotherapy is provided by the recent trials on SGLT-2 inhibitors [34,35,36].

### 4.3. Cluster 2

Cluster 2 seems to be the most interesting. It consists almost exclusively of men. They represent the youngest population with chronic HF with the lowest ejection fraction, developed on aetiology described as “other”. Patients suffered from the least burden of comorbidities, which can be explained by their youngest age and probable underdiagnosis due to low commitment to their health management. These patients can be described as having toxic aetiology. They represent the highest frequency of active smokers and alcohol users and have the highest values of GGTP and bilirubin, which reflect the afflicted liver function [37]. Moreover, they reached the highest mean value of MCV, which might be associated with alcohol abuse [38]. In the clinical assessment, they manifest frequent and massive peripheral oedema, i.e., the highest incidence of lime oedema III, hepatomegaly, and ascites, but somewhat limited pulmonary congestion. This discrepancy between the aggravation of oedema in different vascular areas should be further evaluated. Laboratory signs of congestion, e.g., NT-proBNP, are also the highest among the clusters. Notably, patients in cluster 2 achieved the lowest pCO2, which can be a sign of heightened chemosensitivity—the predictor of an a unfavourable outcome [39].

Notably, the cluster with the lowest incidence of AKI and WRF (cluster 2) was the one with the highest one-year mortality (non-significant). In our opinion, that can be explained by two intertwined hypotheses. First, creatinine is the late marker of kidney function [40] and has limited value in assessing renal damage [41]. Some authors distinguish true and pseudo-WRF based on the concentration of so-called new renal biomarkers, i.e., NGAL, KIM-1, and cystatin-c [42]. Considering this, the isolated increase in serum creatinine can be insufficient for an accurate kidney assessment. Secondly, creatinine can rise during decongestive therapy [43,44]. It was reported that the transient rise of creatinine during decongestive treatment could even be a promising sign, as it reflects the exhaustiveness of the decongestion [45]. Thus, increased creatinine during diuretic treatment does not necessarily indicate genuine kidney injury, which would worsen the outcome, but it can be a sign of diminishing volume overload. The incompleteness of the decongestion was shown to be an important prognostic factor of mortality in AHF [46], which, in our case, could explain why the cluster with the lowest WRF incidence reaches the highest mortality.

The proposed novel classification may complement the classical ways of AHF patient profiling and has significant clinical implications. Each of the extracted clusters has a different suggested pathophysiology and, therefore, another therapeutic pathway that can be therapeutically addressed; e.g., cluster 0-uptitration of the evidence-based HFrEF medical therapy, cluster 1-comorbidities management, and cluster 2-substance abuse counselling and harm reduction. Focusing on these aspects should lead to more accurate treatment tailoring and eventually optimization of therapy. The efficiency of the proposed cluster-based approach to the therapy adjustments should be evaluated in the prospective studies. Notably, clustering does not reveal baffling relationships. The uncovered connections are clear for the experienced cardiologist. The value of the presented analyses is that it provides tangible evidence for the existence of such phenogroups. Potentially, clustering could immediately categorize a patient into one of the groups and suggest to a physician a relevant proceeding, which can sometimes be omitted due to overworking or lack of experience.

### 4.4. Limitations

Our study is not free from limitations. Our data comes from the single-centre registries gathered between 2010–2012 and 2016–2017. Patients in these registries were treated with the current ESC criteria, which did not mention the modern drugs, i.a., a SGLT-2 inhibitor. This influences the potential extrapolation of our results to the present AHF population. Further, we did not assess the novel kidney markers, which would increase the thoroughness of the renal status evaluation. However, the presented assessment model mirrors the commonly used, well-understood variables. Importantly, we have only included the patients who had their creatinine evaluated at four time points, including discharge. Thus, we only included patients who survived the hospitalization. We have also prespecified the number of clusters, as we wanted to avoid the over-fragmentation of the data; however, pre-specification of the number of clusters to three follows the previous papers about clustering in HF [27,47,48]. All the issues mentioned above should be addressed in further trials. 

## 5. Conclusions

Machine learning techniques provided fresh insights into the existing medical datasets. We were able to distinguish three clinically and prognostically different phenotypes. Importantly, these phenotypes are different in terms of the AKI or WRF occurrence. These groups constitute valuable insight into AHF and WRF interplay and may be leveraged for future trial construction and more tailored treatments. Our data provides further evidence for the hypothesis that the serum creatinine concentration should be analysed in the broader context in the population of decongested patients and that its increase is not necessarily prognostically worrying.

Noteworthy, we used the k-medoids algorithm instead of the more popular k-means algorithm because k-medoids represent centroids of clusters as existing data points (patients in our case). This makes the results better interpretable. The k-medoids algorithm is also more robust to outliers than the k-means algorithm [49], which is meaningful in medical data.

## Figures and Tables

**Figure 1 biomolecules-12-01616-f001:**
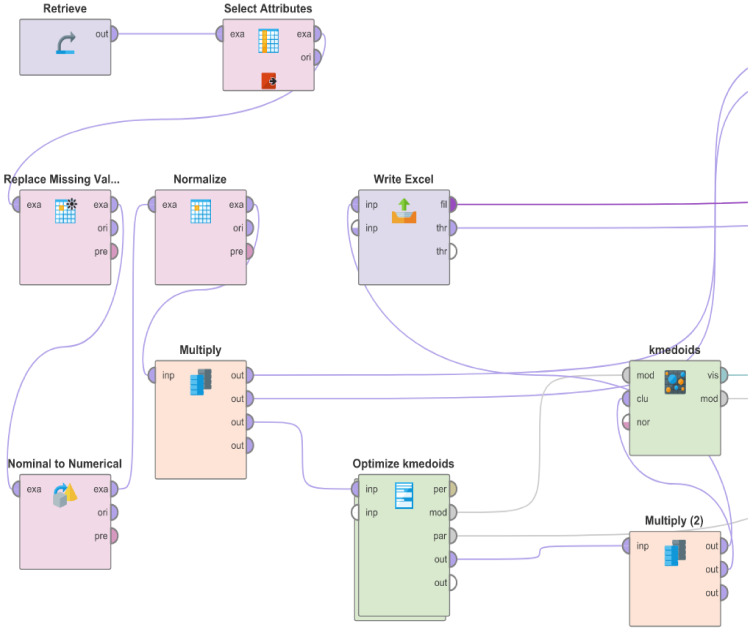
The process of the clusters’ calculation was performed in RapidMiner. The file is attached in the Appendix A.

**Figure 2 biomolecules-12-01616-f002:**
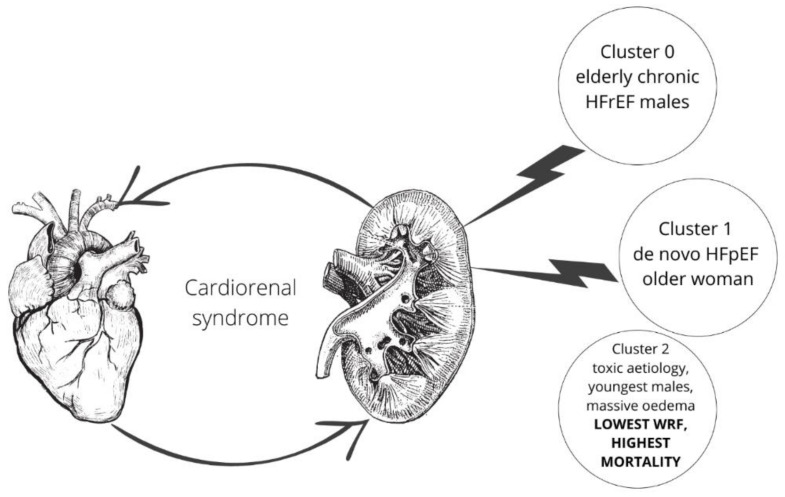
Summary of the most important cluster characteristics and association with renal function.

**Table 1 biomolecules-12-01616-t001:** Variables included in the analysis. All parameters were assessed at admission.

Demographics	Age, Sex
HF characteristics	De novo or chronic HF, Etiology
Comorbidities	Coronary artery disease, myocardial infarction, PCI/CABG, Hypertension, Valvular heart disease, Diabetes, Diabetes treated with insulin, oral drugs or diet, stroke, COPD
Clinical status	Dyspnoea at rest, Dyspnoea at rest (since number of days), NYHA scale at admission, Swelling of the lower limbs, Decrease in exercise tolerance, Decrease in exercise tolerance (since number of days), Body weight, Systolic pressure, Diastolic pressure, Heart rate, Jugular veins pressure, Pulmonary congestion, Pulmonary congestion, Ascites, Hepatomegaly, Implantable device: none = 0, 1-PM, 2-ICD, 3-CRT2
Lifestyle factors	Smoking status (0 = never, 1 = now, 2 = in the past), how many cigarettes did the patient smoke, Active alcohol use, how many cigarettes patients smoke daily, ow many years did/does the patient smoke
Laboratory parameters	PH serum, pCO_2_, pO_2_, ctO_2_, BO_2_, HCO_3_, HCO_3_std, ctCO_2_, BE, sO_2_, FO_2_Hb, FHHb, ctHb, Lac, mOsm, HGB, HCT, RBC, MCV, MCH, MCHC, RDW, WBC, LYMPH, MONO, NEUTR, PLT, Na serum, K serum, Creatinine serum, Urea serum, Glucose serum, Ast, Alt, CRP, GGTP, NTproBNP, Total_bilirubin, INR, Albumins serum, Na urine, K urine, Urea urine, Creatinine urine, Fe, TIBC, Tsat, sTfR, Ferritin, IL-6
Echocardiography	Reduced ejection fraction; ejection fraction

Abbreviations: pCO_2_—partial pressure of CO_2_, pO_2_—partial pressure of O_2_, ctO_2_—concentration of O_2_, BO_2_ -, HCO_3_—bicarbonate, HCO_3_std—bicarbonate standardized, ctCO_2_—CO_2_ concentration, BE—base excess, sO_2_—O_2_ saturation, FO_2_Hb—fraction of oxygenated haemoglobin, FHHb—fraction of deoxyhemoglobin in total hemoglobin, ctHb—total hemoglobin, Lac—lactates, mOsm –milliosmoles, HGB—hemoglobin, HCT—hematocrit, RBC—red blood count, MCV—mean corpuscular volume, MCH—mean corpuscular hemoglobin, MCHC—mean corpuscular hemoglobin concentration, RDW—red cell distribution width, WBC—white blood count, LYMPH—lymphocytes percentage, MONO—monocytes, NEUTR—neutrophiles, PLT—platelets count, Ast—aspartate aminotransferase, Alt—alanine transaminase, CRP—C-reactive protein, GGTP—gamma-glutamyl transpeptidase, NTproBNP—N-terminal prohormone of brain natriuretic peptide, INR—international normalized ratio, Fe—total iron amount in blood, TIBC—total iron-binding capacity, Tsat—transferrin saturation, sTfR—Soluble Transferrin Receptor, IL-6—interleukin 6th, eGFR—estimated glomerular filtration rate.

**Table 2 biomolecules-12-01616-t002:** Characteristics of patients in each cluster and the whole group. The lowest values are marked green and the highest ones are red.

Parameter	Cluster 0	Cluster 1	Cluster 2	Global	*p*
Demographics					
*n*	158 (51%)	110 (35%)	44 (14%)	312	
Sex, male (*n*)	138 (87%)	53 (48%)	43 (98%)	234 (75%)	<0.001
Age (years)	69.192 ± 11.826	72.217 ± 11.736	56.015 ± 13.273	68.4 ± 13.054	<0.001
AHF characteristics					
Ejection fraction	30 (25–38.5)	43 (30–53.5)	25 (15–35)	32 (25–45)	<0.001
Chronic HF (*n*)	133 (84%)	29 (26%)	34 (77%)	196 (63%)	<0.001
Reduced EF (*n*)	133 (84%)	59 (54%)	34 (77%)	226 (72%)	<0.001
Aetiology					<0.001
Coronary artery disease (*n*)	120 (76%)	19 (17%)	8 (18%)	147 (47%)	
Valvular (*n*)	12 (8%)	21 (19%)	6 (14%)	39 (13%)	
Hypertension (*n*)	5 (3%)	6 (5%)	0 (0%)	11 (4%)	
Other (*n)*	21 (13%)	64 (58%)	30 (68%)	115 (37%)	
Comorbidities					
Coronary artery disease (*n*)	139 (88%)	37 (34%)	7 (16%)	183 (59%)	<0.001
Myocardial infarction in the past (*n*)	73 (46%)	22 (20%)	4 (9%)	99 (32%)	<0.001
PCI/CABG in the past (*n*)	75 (47%)	19 (17%)	3 (7%)	97 (31%)	<0.001
Hypertension (*n*)	127 (80%)	100 (91%)	10 (23%)	237 (76%)	<0.001
Valvular disease (*n*)	113 (72%)	62 (56%)	29 (66%)	204 (65%)	0.037
Diabetes mellitus (*n*)	63 (40%)	56 (51%)	5 (11%)	124 (40%)	<0.001
Diabetes treatment (*n*)					0.002
Insulin	25 (16%)	11 (10%)	1 (2%)	37 (12%)	
Oral drugs	28 (18%)	25 (23%)	4 (9%)	57 (18%)	
Diet	6 (4%)	4 (4%)	0 (0%)	10 (3%)	
Stroke (*n*)	21 (13%)	14 (13%)	6 (14%)	41 (13%)	0.986
COPD (*n*)	27 (17%)	9 (8%)	4 (9%)	40 (13%)	0.073
Clinical status					
Dyspnoea at rest (*n*)	131 (83%)	84 (76%)	35 (80%)	250 (80%)	0.299
Dyspnoea at rest lasts for (number) days	3 (2–7)	2 (1–7)	5.5 (2.5–8.5)	3 (1–7)	0.370
Deterioration of effort tolerance (*n*)	152 (96%)	103 (94%)	39 (89%)	294 (94%)	0.175
Deterioration of effort tolerance (number) days	14 (7–21)	14 (7–30)	14 (7–30)	14 (7–28)	0.021
NYHA (n)					<0.001
I	5 (3%)	3 (3%)	4 (9%)	12 (4%)	
II	39 (25%)	8 (7%)	9 (20%)	56 (18%)	
III	42 (27%)	19 (17%)	14 (32%)	75 (24%)	
IV	64 (41%)	62 (56%)	14 (32%)	140 (45%)	
Swelling of lower limbs (*n*)					0.050
Swelling of lower limbs 0	43 (27%)	32 (29%)	8 (18%)	83 (27%)	
Swelling of lower limbs 1	37 (23%)	25 (23%)	8 (18%)	70 (22%)	
Swelling of lower limbs 2	50 (32%)	32 (29%)	12 (27%)	94 (30%)	
Swelling of lower limbs 3	28 (18%)	20 (18%)	16 (36%)	64 (21%)	
JVP (n)					0.005
JVP 1	97 (61%)	70 (64%)	17 (39%)	184 (59%)	
JVP 2	51 (32%)	38 (35%)	23 (52%)	112 (36%)	
JVP 3	10 (6%)	1 (1%)	4 (9%)	15 (5%)	
Pulmonary congestion (*n*)	147 (93%)	91 (83%)	40 (91%)	278 (89%)	0.026
Pulmonary oedema (*n*)					0.108
no	11 (7%)	18 (16%)	4 (9%)	33 (11%)	
up to 1/3 of lungs	102 (65%)	48 (44%)	33 (75%)	183 (59%)	
up to 2/3	35 (22%)	24 (22%)	5 (11%)	64 (21%)	
>2/3	10 (6%)	19 (17%)	2 (5%)	31 (10%)	
Ascites (*n*)	19 (12%)	7 (6%)	15 (34%)	41 (13%)	<0.001
Hepatomegaly (*n*)	25 (16%)	14 (13%)	26 (59%)	65 (21%)	<0.001
Implantable device (*n*)					<0.001
PM	16 (10%)	7 (6%)	1 (2%)	24 (8%)	
ICD	43 (27%)	2 (2%)	3 (7%)	48 (15%)	
CRT	15 (9%)	2 (2%)	4 (9%)	21 (7%)	
Systolic pressure (mmHg)	130 (110–150)	145 (124–171)	110 (100–127)	130 (110–150)	<0.001
Diastolic pressure (mmHg)	75.5 (70–87)	83 (70–100)	70 (60–83.5)	80 (70–90)	<0.001
Heart rate (beats per minute)	78 (70–100)	90 (72–110)	100 (80–110)	83 (70–100)	<0.001
Body weight (kg)	80 (72.55–93)	78.25 (68.5–88.6)	76 (67–87.8)	79.2 (70–91)	0.437
Lifestyle factors					
Smoking status (*n*)					<0.001
Never	74 (47%)	74 (67%)	15 (34%)	163 (52%)	
Active	18 (11%)	15 (14%)	15 (34%)	48 (15%)	
In the past	66 (42%)	21 (19%)	14 (32%)	101 (32%)	
How many cigarettes patient smoke daily (*n*)	10 (0–20)	10 (0–20)	10 (0–20)	10 (0–20)	0.797
How many years did the patient smoke/does the patient smoke cigarettes (*n*)	20 (0–30)	10 (0–30)	13 (0–30)	20 (0–30)	0.380
Active alcohol use (*n*)	40 (25%)	18 (16%)	29 (66%)	87 (28%)	<0.001
Laboratory parameters					
HGB (g/dL)	13.232 ± 1.993	12.955 ± 1.892	13.975 ± 1.759	13.239 ± 1.947	0.013
HCT (%)	39.844 ± 5.535	39.145 ± 5.233	41.766 ± 5.173	39.868 ± 5.427	0.025
RBC (× 1012/L)	4.482 ± 0.663	4.389 ± 0.568	4.552 ± 0.586	4.459 ± 0.621	0.274
MCV (fL)	89.195 ± 6	89.31 ± 5.743	92.125 ± 7.124	89.649 ± 6.145	0.015
MCH (pg)	29.597 ± 2.253	29.546 ± 2.466	30.828 ± 2.992	29.749 ± 2.472	0.008
WBC (× 109/L)	8.54 (6.5–10.3)	8.1 (6.5–10.4)	8.45 (7.1–9.85)	8.3 (6.6–10.3)	0.872
PLT (× 109/L)	196 (159–242)	201 (158–248)	207 (174–250)	198 (159–245)	0.777
pH	7.45 (7.42–7.48)	7.425 (7.375–7.465)	7.45 (7.43–7.49)	7.44 (7.41–7.47)	0.003
sO_2_ (%)	92.85 (90.1–95.45)	93.55 (91.3–94.9)	93.8 (88.7–96.3)	93.2 (90.4–95.4)	0.946
pO_2_ (mmHg)	65.35 (57.9–73.25)	67.4 (62.4–74.45)	66.9 (55.2–80.6)	66.1 (59–74.6)	0.956
pCO_2_ (mmHg)	34.65 (32.15–38.8)	35.8 (32.4–39.25)	33.4 (30.4–36.9)	35.2 (32–38.9)	0.517
HCO_3_ (mmol/L)	24.025 ± 3.223	22.805 ± 3.416	23.939 ± 4.62	23.578 ± 3.558	0.025
BE mEq/l	0.197 ± 3.42	-1.252 ± 3.712	0.3 ± 4.523	-0.304 ± 3.755	0.007
mOsm (Osm/L)	281 (274–286)	286 (280–290)	274 (264-285)	282 (274–288)	<0.001
Na (mmol/L)	139 (137–142)	140 (138–142)	136.5 (133.5–141)	139 (136–142)	0.007
K (mmol/L)	4.117 ± 0.549	4.296 ± 0.627	4.107 ± 0.537	4.179 ± 0.581	0.031
Lactates (mmol/L)	1.9 (1.5–2.4)	2 (1.5–2.7)	2 (1.6–3)	2 (1.5–2.55)	0.088
Glucose (mg/dL)	120 (102–157)	131 (106–186)	107.5 (94.5–126.5)	120 (102.5–152)	0.001
INR	1.37 (1.12–1.8)	1.32 (1.12–1.93)	1.35 (1.175–1.7)	1.345 (1.12–1.8)	0.102
Albumin (g/dL)	3.775 ± 0.367	3.743 ± 0.402	3.602 ± 0.42	3.738 ± 0.39	0.036
Ast (IU/L)	26 (19–34.5)	29.5 (22–41)	28.5 (20.5–40)	27 (20–38)	0.004
Alt (IU/L)	28 (17.5–47)	31 (20.5–55.5)	28 (18.5–44)	29 (19–48)	0.019
Total bilirubin (mg/dL)	1.06 (0.77–1.6)	0.9 (0.63–1.47)	1.415 (0.905–2.455)	1.04 (0.72–1.67)	<0.001
GGTP (IU/L)	70 (45–135)	54.5 (29–103)	99 (48–206)	69.5 (40–123.5)	0.021
CRP (mg/L)	7.04 (4–15.4)	6.2 (2.6–14)	10.25 (4.35–24.35)	7.1 (3.4–16.2)	0.513
IL6 (pg/mL)	8.346 (1.155–21.1)	11.705 (3.257–26.299)	11.352 (6.338–30.117)	10.056 (2.508–22.9)	0.734
Ferritin (ng/mL)	101.9 (51.94–191)	125.6 (65.5–218.75)	115.15 (51–287.9)	105.7 (57.08–212)	0.372
Tsat (%)	16.1 (12.3–21.9)	15.608 (11.9–18.519)	16.026 (10.7–26.6)	15.84 (11.82–21.1)	0.030
sTfR (mg/L)	1.885 (1.495–2.46)	1.82 (1.46–2.46)	1.755 (1.4–2.53)	1.85 (1.46–2.46)	0.972
TIBC (μg/dL)	352.192 ± 72.92	338.052 ± 62.645	366.357 ± 76.825	349.514 ± 70.639	0.075
Fe (μg/dL)	56 (43–79)	51 (40–64)	60.5 (43–88)	54 (42–73)	0.005
NTproBNP (pg/mL)	5291 (3081–9203)	5525 (2755–13,629)	7106 (5026–11,759)	5659 (3119–10,572)	0.021
Creatinine (mg/dL)	1.21 (1.04–1.47)	1.23 (0.95–1.62)	1.14 (0.925–1.44)	1.21 (1.005–1.49)	0.761
Urine Creatinine (mg/dL)	69.4 (37.1–126.5)	43.5 (27.6–88.7)	73.6 (34.7–125.9)	61.5 (31.6–110.9)	0.026
Urea (mmol/L)	52 (39–73)	48 (38–73)	56 (39–74)	51 (38–73)	0.224
Urine Urea (mmol/L)	841 (506–1413)	581 (384–1232)	1122.5 (482–1663)	813 (433–1437)	0.023
Urine K (mmol/L)	30 (20.53–43.27)	26 (17–39)	28.415 (19.6–45)	29 (19–42.59)	0.249
Urine Na (mmol/L)	88.253 ± 38.623	97.721 ± 32.398	84.786 ± 48.268	91.05 ± 38.333	0.078

**Table 3 biomolecules-12-01616-t003:** Key clinical features of each cluster.

Cluster	Key Clinical Features
**Cluster 0**	Most numerous cluster. Highest: % of chronic and reduced EF HF, CAD, Valvular heart disease, COPD, implanted electric devices, pulmonary congestion, albumins, HCO_3_, Tsat, insulin-dependent, and diet-treated diabetes. Lowest: deterioration of effort tolerance (number) of days, HR, MCV, Ast, Alt, NT-proBNP. Non-significant: highest % of dyspnea at rest, deterioration of effort tolerance, swelling of the lower limbs 1, 2, body weight, past smokers. Lowest: limbs oedema III, JVP II, active smokers—**elderly chronic HFrEF male, with mild congestion, moderate WRF and AKI, and one-year mortality occurrence**
**Cluster 1**	Highest: % of females, age, ejection fraction, % of de novo HF and preserved EF, valvular and hypertension aetiology, hypertension, diabetes, RR, mOsm, Na, K, glucose, Ast, Alt, lowest: ascites, hepatomegaly, HGB, HCT, MCH, pH HCO_3_, urine creatinine and urea, Non-significant: highest: NYHA IV, limbs oedema I, JVP I, no pulmonary oedema, pCO_2_, IL-6, ferritin, creatinine, urine Na—**first manifestation of HFpEF older woman, with high inflammatory markers, creatinine and osmolarity, highest AKI and WRF occurrence, and moderate one-year mortality**
**Cluster 2**	Highest: % of males, other aetiology, stroke history, ascites, hepatomegaly, HR, active alcohol users, HGB, HCT, MCV, bilirubin, GGTP, Fe, NT-proBNP, and urine creatinine and urea. Lowest: age, ejection fraction, CAD history, RR, mOsm, Na, K, glucose, and albumin Non-significant: highest: active smokers, limbs oedema III, pulmonary oedema I. Lowest: body weight, CO_2_, creatinine, urine Na—**young men, with massive oedema and substance abuse involvement, low AKI and WRF occurrence, and highest one-year mortality**

Abbreviations: EF—ejection fraction, HF—heart failure, CAD—coronary artery disease, COPD—chronic obstructive pulmonary disease, Tsat—transferrin saturation, HR—heart rate, MCV—mean corpuscular volume, Ast—aspartate aminotransferase, Alt—alanine transaminase, NT-proBNP—N-terminal brain natriuretic peptide, JVP—jugular venous pressure, HFrEF—heart failure with reduced ejection fraction, WRF—worsening of renal function, AKI—acute kidney injury, RR—blood pressure, mOsm—osmolarity, N—sodium, K—potassium, HGB—haemoglobin, HCT—haematocrit, MCH—mean corpuscular haemoglobin, NYHA—New York Heart Association scale, IL-6—interleukin 6, GGTP—gamma-glutamyl transferase, and Fe—iron.

**Table 4 biomolecules-12-01616-t004:** Outcome by cluster and in the whole group.

Parameter	Cluster 0	Cluster 1	Cluster 2	Global	*p*
WRF, *n*	24 (15%)	26 (24%)	1 (2%)	51 (16%)	0.004
AKI, *n*	12 (8%)	17 (15%)	0 (0%)	29 (9%)	0.007
Time of hospitalization (days)	6 (5–9)	7 (5–9)	8 (6–14)	7 (5–9.5)	0.006
In hospital deterioration of HF, *n*	9 (6%)	7 (6%)	3 (7%)	19 (6%)	0.856
One year mortality, *n*	35 (22%)	24 (22%)	15 (34%)	74 (24%)	0.200

Abbreviations: WRF—worsening of the renal function, AKI—acute kidney injury, HF—heart failure.

## Data Availability

The data presented in this study are available within the article. Further data are available on request from the corresponding author.

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
