# Peer review of "Machine Learning Approach to Understand Worsening Renal Function in Acute Heart Failure"

_biomolecules, 2022, doi:10.3390/biom12111616_

Round 1

Reviewer 1 Report

The rationale for choosing The K-medoids algorithm was not mentioned or clear. The research gap the paper is going to address is not clear. 

The conclusion:

Machine learning techniques provided fresh insight into the existing medical da- 290 tasets. We were able to distinguish 3 clinically and prognostically different phenotypes. 291 Importantly, these phenotypes are different in terms of the AKI or WRF occurrence. These 292 groups constitute valuable insight into AHF and WRF interplay and may be leveraged for 293 future trial construction and more tailored treatment. Our data brings further evidence 294 for the hypothesis that the serum creatinine concentration should be analyzed in the 295 broader context in the population of decongested patients and that its increase is not nec- 296 essarily prognostically worrying

Needs revision the authors has to present their conclusion on the specific algorithm used it looks generic in the present form.

Reviewer 2 Report

Authors present a Machine learning approach to understand worsening renal 2

function in acute heart failure. Authors are suggested to enlist main contributions at the end of the introduction section. Reproducibility of the study is important for reliability of the work and also for future research and improvements. Therefore, authors are suggested to provide computer code/programmes developed and used for this study in some publicly open repository like GitHub/ResearchGate or some other open source storage, and provide the link for code and data in the paper. Paper is well written and can be accepted for publication with the above changes.
